# Spatial Differentiation of Land Use and Landscape Pattern Changes in the Beijing–Tianjin–Hebei Area

**Yafei Wang [1,2], Xiaoli Zhao [1,*], Lijun Zuo [1], Zengxiang Zhang [1], Xiao Wang [1], Ling Yi [1], Fang Liu [1] and Jinyong Xu [1]**

[1]  Aerospace Information Research Institute, Chinese Academy of Sciences, Beijing 100094, China; wangyf@radi.ac.cn (Y.W.); zuolj@radi.ac.cn (L.Z.); zhangzx@radi.ac.cn (Z.Z.); wangxiao98@radi.ac.cn (X.W.); Yiling@radi.ac.cn (L.Y.); liufang@radi.ac.cn (F.L.); xujy@radi.ac.cn (J.X.)

[2]  University of Chinese Academy of Sciences, Beijing 100049, China

*  Correspondence: zhaoxl@radi.ac.cn; Tel.: +86-1368-363-7349

**Abstract:** Landscape pattern analysis based on geometric features effectively reflects the spatial patterns of land use. Based on the administrative boundaries of prefecture-level cities, the Beijing–Tianjin–Hebei collaborative development area is divided into three sub-regions, according to ecological–production–living functions. We used remote sensing data of long time series land-use change from late 1980 to 2015, and analyzed landscape pattern changes and spatial differentiation in the past 30 years. The results show that: (1) The main type of land-use change was the flow of cultivated land to urban construction land, and the urbanization process was significant. (2) The urban construction land was the landscape type with the highest degree of fragmentation and maximum land-use change in the region. (3) The patch density in the Beijing–Tianjin–Hebei area increased while the average patch area decreased, and the entire landscape tended towards significant fragmentation. The Shannon diversity and evenness indexes continued to increase, indicating that the overall landscape in this region is heterogeneous and diversified. The ecological and environmental protection measures implemented in this region so far have achieved results, but require more stringent measures to ensure the total diversification of land use in the region.

**Keywords:** Beijing–Tianjin–Hebei collaborative development area; land use; landscape pattern; spatial and temporal characteristics; ecological-production-living space

## 1. Introduction

In ecology, a landscape pattern is defined as the arrangement of patches with different sizes and shapes in a landscape space according to certain laws [1]. Landscape pattern analysis aims to describe the status quo of a landscape and explain the process of its emergence [2], and the way of land use, which plays a major driving role in the change of landscape patterns [3]. Therefore, the combination of land use change and landscape pattern is an effective method to reveal the regional ecological environment and its spatial differentiation characteristics [4]. At present, landscape ecology is in a vigorous development stage, and studies on regional landscape patterns are becoming increasingly valued [5]. To reveal the temporal and spatial distribution law of the evolution of a landscape pattern for a certain area or a certain land cover type (such as forest land and grassland), scholars have conducted extensive research on this aspect. The relevant results show that the degree of landscape fragmentation in a certain area is positively correlated to its urbanization level, and its fragmentation trend is closely related to human interference activities in the past few decades [6,7]. An increase in construction leads to the overall fragmentation of regional ecological landscapes. Most notably, cultivated landscapes are

rapidly declining with the rise in construction [8], and several ecological landscapes are also showing a degrading trend as a result [9].

The coordinated development of the Beijing–Tianjin–Hebei area is a major national strategy in China. As one of China's three major economic growth poles, this area holds 8% of the country's population with 2.3% of the country's land, and creates 11% of the country's GDP. Therefore, the coordinated development strategy of the Beijing-Tianjin-Hebei area plays an irreplaceable role in reshaping China's economic territory. Regional integration has effectively promoted the economic growth of the urban agglomeration, but at the same time, several prominent issues have emerged, such as the unbalanced development of the urban system, continuous deterioration of the ecological environment and the increasingly acute contradiction between human and land [10]. Understanding the landscape pattern changes in the Beijing–Tianjin–Hebei area is the premise for minimizing the negative impacts of construction on the ecological environment and ensuring sustainable development. Most of the studies on landscape patterns have focused on the Beijing–Tianjin–Hebei area as a whole without considering the main functions of each of the different cities [11,12]. Therefore, this paper attempts to classify the cities in the Beijing–Tianjin–Hebei area into three sub-regions, namely the Agricultural Development Region, the Beijing–Tianjin Urban Agglomeration Region and the Northwest Water Conservation Region, according to their production, living and ecological functions. Based on this classification, this study analyzes the spatial differentiation law of land use and landscape pattern change in this area in the last 30 years to provide a reference and basis for optimizing the spatial pattern of ecological–production–living and further promote the healthy and sustainable development of the Beijing–Tianjin–Hebei area [13].

## 2. Materials and Methods

### 2.1. Data Overview

The Beijing–Tianjin–Hebei collaborative development area is located between 113°27′–119°50′E and 36°05′–42°40′N, and includes Beijing, Tianjin, and Hebei Province. It covers an area of approximately $2.16 \times 10^5$ km², and accounts for 2.3 % of the national territory. This area is dominated by plain terrain, showing the topographic characteristics of high northwest and low southeast. In 2015, with a GDP of 6.65 trillion yuan and a population of $1.11 \times 10^8$, it became the most economically developed region in north China.

Based on China's 1:100000 scale long time series land use database, constructed by the Institute of Remote Sensing and Digital Earth, Chinese Academy of Sciences, using GIS technology, we obtained the land use data for the Beijing–Tianjin–Hebei area in the late 1980s, 2000, 2005, 2010, and 2015. The land use remote sensing monitoring classification system includes six primary classes and 27 secondary classes (Figure 1). The first level is classified as follows: cultivated land, forest land, grassland, water area, construction land, and unused land [14]. The projection adopted is the double standard weft equal-area secant conic projection, with the national unified central longitude and double standard weft. The central longitude is 105°E, and the double standard weft is 25°N and 47°N. KRASOVSKY ellipsoid was used for this study.

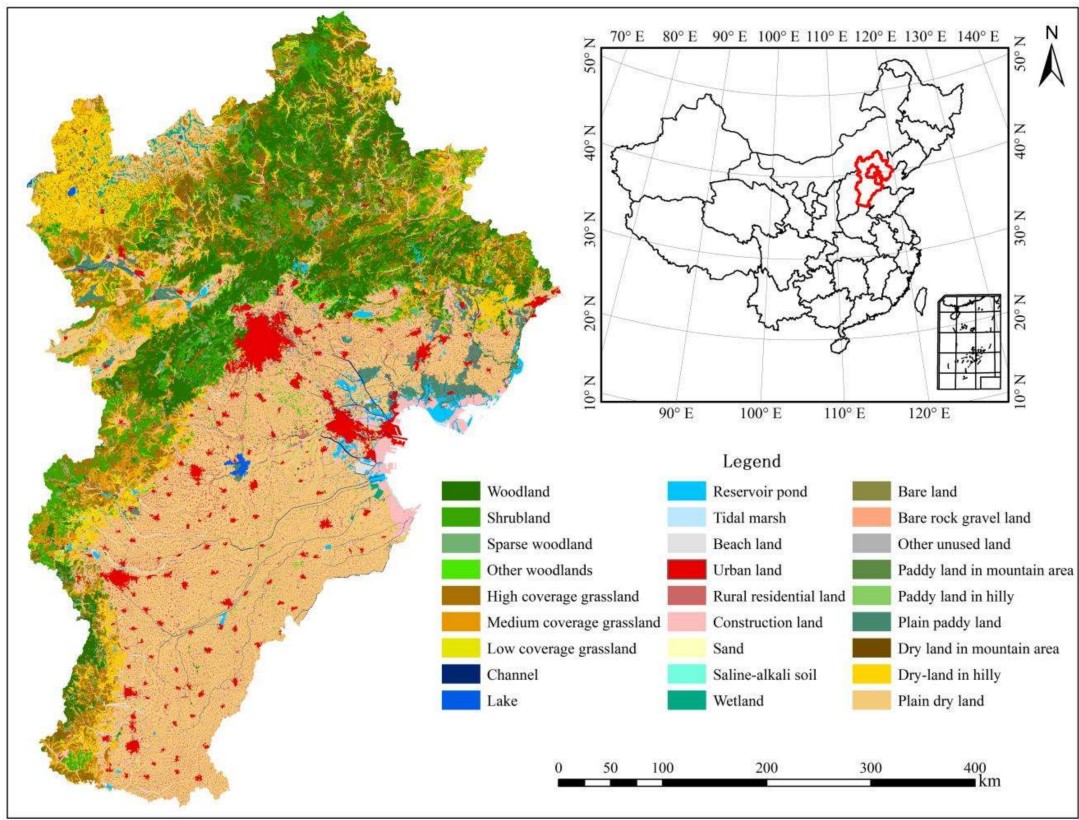

**Figure 1.** Land use in the Beijing–Tianjin–Hebei collaborative development area in 2015.

## 2.2. Research Methods

### 2.2.1. Analysis of Dynamic Changes in Land Use

In this study, the spatial and temporal changes of land use over the past 30 years are studied by combining the dynamic degree of land use and the transformation matrix of land-use types, and the dynamic characteristics and causes of different land-use types are emphatically analyzed.

The comprehensive dynamic degree of land use, which refers to the quantitative change in land use in a certain time range in a specific research area, is expressed as follows [15]:

$$C = (\frac{\sum_{i=1}^{n} \Delta S_{i-j}}{2\sum_{i=1}^{n} S_i})/t \times 100\% \tag{1}$$

where $C$ is the comprehensive dynamic degree of land-use type; $S_i$ is the area of class $i$ land-use type at the initial time of the study; $\Delta S_{i-j}$ is the absolute value of the area in which class $i$ converted to class $j$ in the research period, $i \neq j$; $n$ is the number of land-use types in the area, and $t$ is the length of the monitoring period.

### 2.2.2. Landscape Pattern Analysis

Analysis of landscape patterns aims to describe the current situation of the landscape and attempts to explain the process of its emergence. The landscape index can highly summarize landscape pattern information and reflects characteristics of temporal and spatial changes [1]. In this study, ArcGIS software was used to convert the first-level land-use type map into a grid format, and FRAGSTATS (a landscape pattern analysis software) was used to calculate the relevant indicators of the class and

landscape level. Finally, combining spatial distribution characteristics, we analyzed the landscape pattern of land use in the study area.

Given the high correlation between several of the indicators [16], the Patch Density (PD), Average Patch Area (AREA_MN), Landscape Shape Index (LSI), Perimeter–Area Fractal Dimension Index (PAFRAC) and Aggregation Index (AI) were selected to describe the area and shape characteristics of the patches [17–27]; on this basis, the CONTAG, DIVISION, Shannon Diversity Index (SHDI) and Shannon Evenness Index (SHEI) were also selected to reflect the connectivity, fragmentation, diversity, and heterogeneity of the landscape level [18,28]. The indicators involved in this article were calculated in Fragstats using a general method. For the detailed calculation formulas of the indicators, please refer to the description manual of Fragstats 4.2.

## 3. Results

### 3.1. Land-Use Change and its Regional Differences in the Beijing–Tianjin–Hebei Collaborative Development Area

#### 3.1.1. Dynamic Changes in Land Use in the Beijing–Tianjin–Hebei Area

From the late 1980 to 2015, a total of 15,547.06km$^2$ of land in the study area experienced land-use changes, and the comprehensive land use dynamic degree increased from 0.09% in the 1980–2000 period to 0.15 % between 2010–2015 (Figure 2). There were significant differences in the changes of land-use types: construction land (urban and rural industrial land including mining land and residential land) had the largest net change area, with a net increase of 9527.90km$^2$; cultivated land had the largest net decrease in area, with a net decrease of 5217.00km$^2$; further, the areas of grassland, forest land, water area, and unused land were all reduced. The temporal-spatial characteristics of land use can be summarized as having multiple dynamic change types, complex changes, wide distributions, large scales and so on. The main trend, however, observed during this time, was the decrease in cultivated land and the continuous increase in construction.

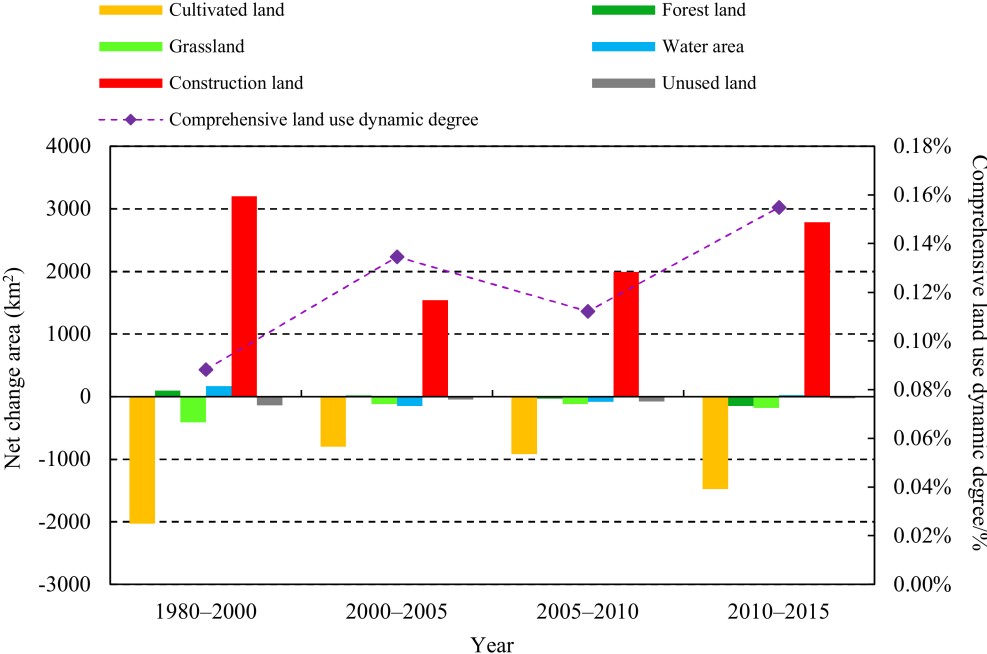

**Figure 2.** Net change area of land-use types and the degree of comprehensive land-use dynamics in the Beijing–Tianjin–Hebei collaborative development area.

### 3.1.2. Regional Differences in Land Use in the Beijing–Tianjin–Hebei Area

*The National Land Planning Outline (2016–2030)* divides the land space into three types namely urban, agricultural, and ecological space [19]. Since the 18th National Congress of the Communist Party of China (CPC), the optimization of the spatial layout consisting of production, living and ecological land has become a hot topic for domestic scholars. Based on this, the Beijing–Tianjin–Hebei coordinated development area is divided into three functional regions for the first time, according to the rule of ecological–production–living space (Figure 3). Firstly, considering that forest land, grassland, water area, and unused land all have water conservation functions, they are combined into ecological land, thereby the six first-class land categories are turned into three categories: cultivated land, urban and rural industrial and mining residential land, and ecological land. The proportion of each land-use type in the total area of the three different regions is then calculated respectively. Finally, the distribution of production, living and ecological functional areas based on the municipal administrative units is obtained.

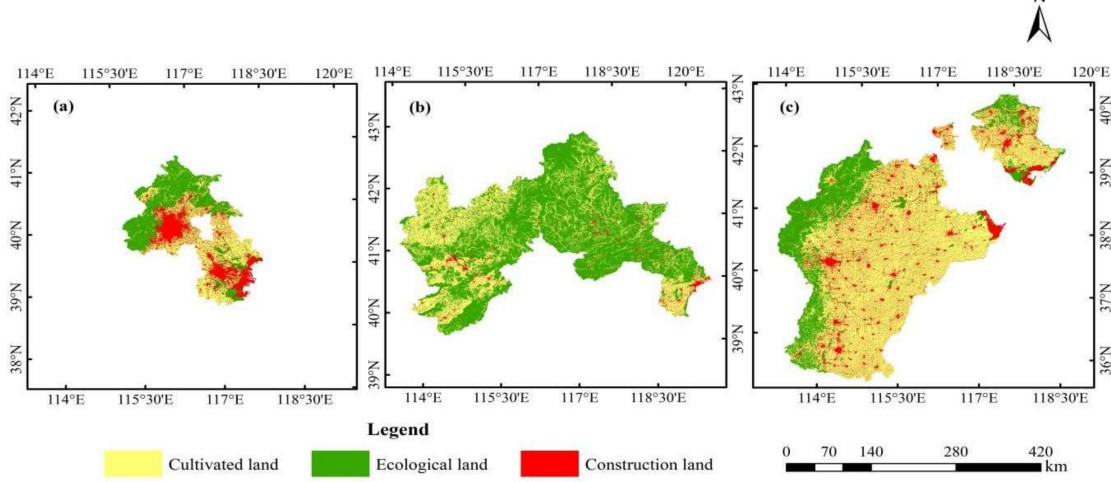

**Figure 3.** Ecological, production, and living functional regions: **(a)** Beijing–Tianjin Urban Agglomeration Region; **(b)** Northwest Water Conservation Region; **(c)** Agricultural Development Region.

The ecological functional region consists of the cities of Zhangjiakou, Chengde and Qinhuangdao, and the main land-use type for these cities is ecological land, accounting for 62.84 % of the total area. Because of these characteristics, it is home to the Northwest Water Conservation Region of the research area. As of 2015, the construction land area of the Beijing–Tianjin Urban Agglomeration Region has increased by 127.67 % compared to the 1980s, and the urbanization rates in Beijing and Tianjin are currently at 86.4 % and 82.3 % respectively. These two cities are mainly responsible for fulfilling the living needs of the population, such as residence, consumption, leisure, and entertainment. Together, they have become the living functional part of the Beijing–Tianjin–Hebei area. The remaining eight cities are made up of predominately cultivated land, accounting for 62.55 % of the Agricultural Development Region. This region is the 'granary' of the Beijing–Tianjin–Hebei area, mainly providing agricultural products and serving as a production functional zone.

The land-use dynamic map of the Beijing–Tianjin–Hebei area was cut according to the range of the three sub-regions, and the land-use dynamics in the three functional zones were obtained. The comprehensive dynamic degrees were respectively calculated based on this (Figure 4).

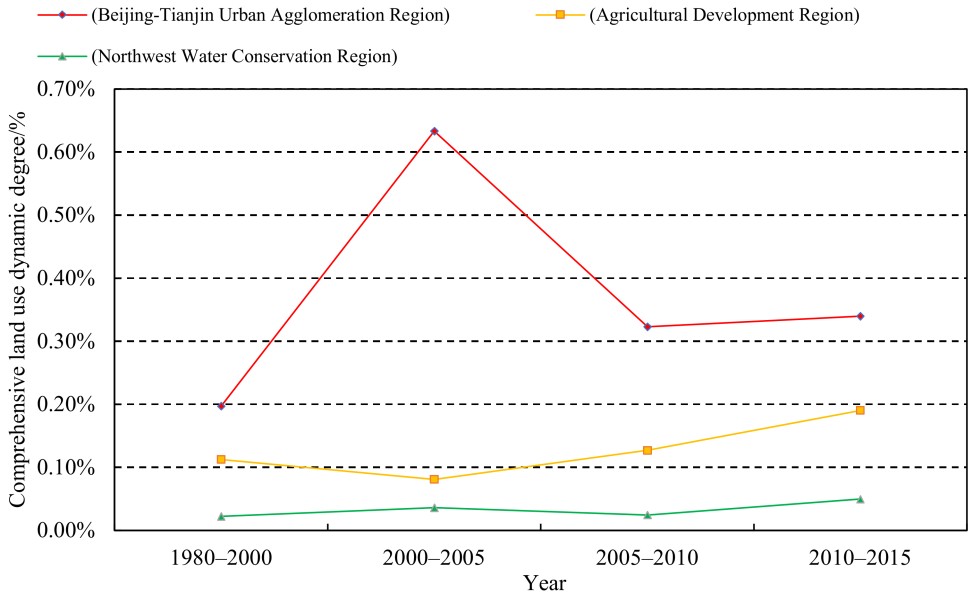

**Figure 4.** Comprehensive dynamic degree of land use in the three regions.

The results show that the dynamic degree of land use in the three functional zones at the end of the study period is higher than that of at the beginning, indicating an overall upward trend. Among them, the Beijing–Tianjin Urban Agglomeration Region has always maintained the highest dynamic degree of land use, with a peak value of 0.63% appearing in the 2000–2005 period. This signifies the following inferences: the dynamic change of land use in this region is more intense when compared with the other two regions; the comprehensive dynamic degree of land use in the Northwest Water Conservation Region is the lowest, and the degree of change is the smallest, decreasing from 0.02% in the 1980–2000 period to 0.05% in between 2010–2015, indicating that the land use here is relatively stable; the comprehensive dynamic degree of land use in the Agricultural Development Region first declined by a small margin and then continued to rise, reaching a peak value of 0.18% between 2010–2015, ranking at a medium level among the three regions.

To assess the similarities and differences between land use transformation types in different functional zones in more detail, we calculated the transfer matrix of cultivated land, ecological land and construction land in the three regions. The results are shown in Figure 5.

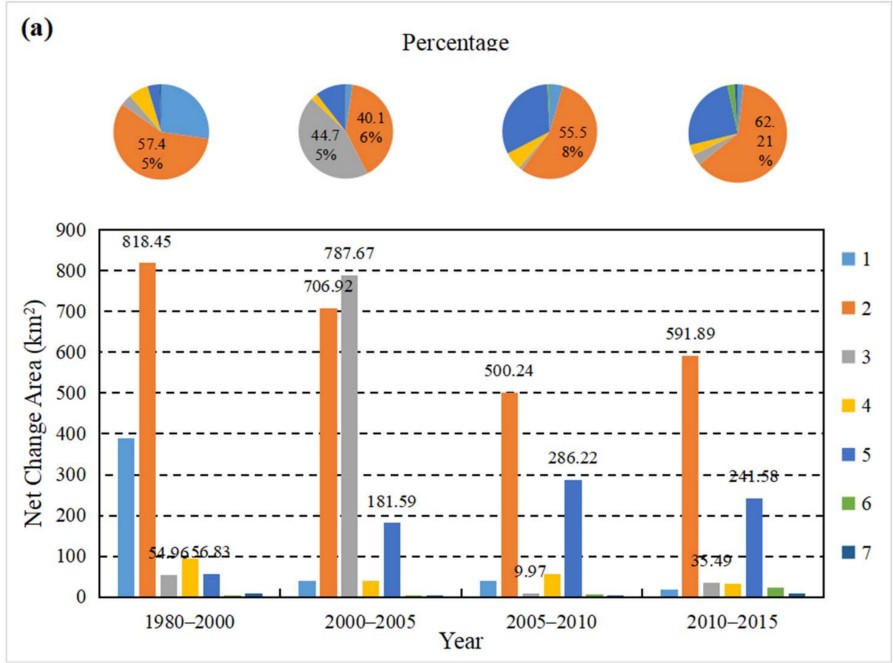

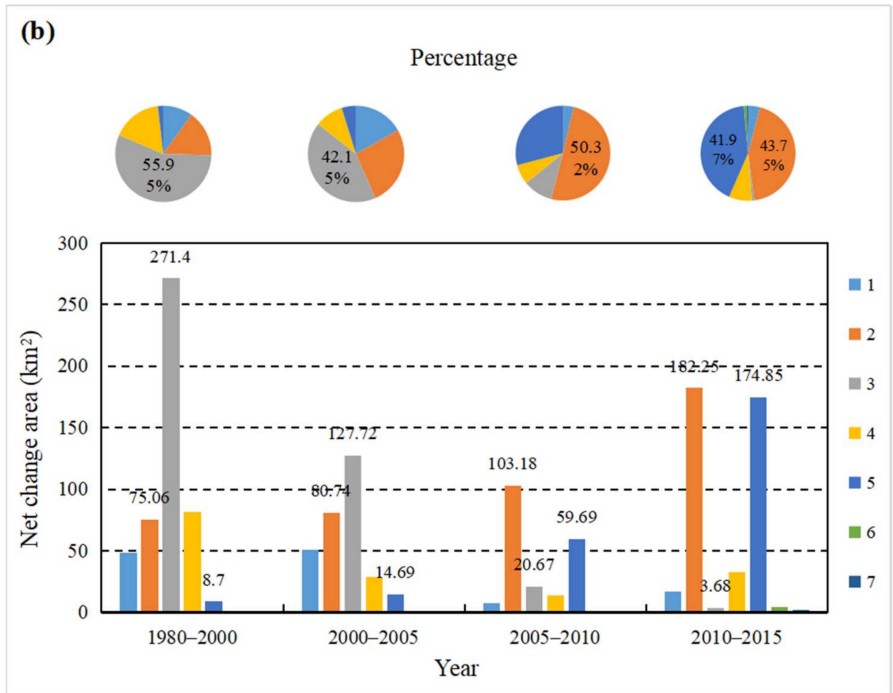

**Figure 5.** *Cont.*

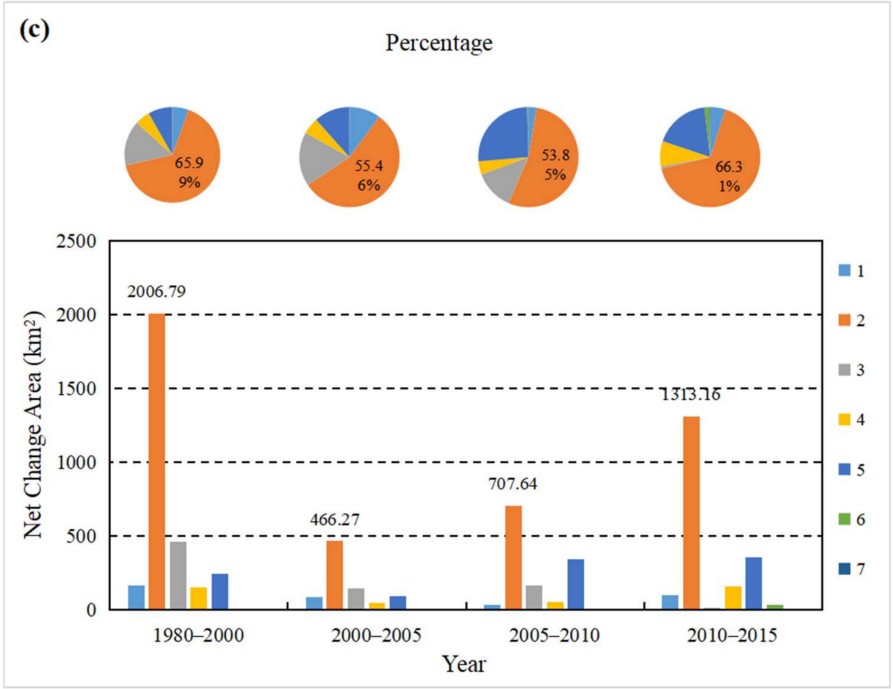

**Figure 5.** Land use dynamic change during each period in the three regions: **(a)** Beijing–Tianjin Urban Agglomeration Region; **(b)** Northwest Water Conservation Region; **(c)** Agricultural Development Region.

Explanations of land use transformation types:

1.  cultivated land → ecological land
2.  cultivated land → construction land
3.  ecological land → cultivated land
4.  ecological land → ecological land
5.  ecological land → construction land
6.  construction land → cultivated land
7.  construction land → ecological land

On the whole, the phenomenon of construction land encroaching on cultivated land is the most commonly occurring, which shows that in the process of urbanization in this area, urban expansion mainly takes up cultivated land. The characteristics of land use change in the living area and production area are relatively consistent. The main transfer type is the flow of cultivated land to construction land, and its transfer area accounts for more than 60% in the Beijing–Tianjin Urban Agglomeration Region. During the 1980–2000 period, 2006.79 $km^2$ of agricultural land in the Agricultural Development Region was developed into urban construction land. This number decreased to 1313.16 $km^2$ in the 2010–2015 period, but still accounted for 66.31% of the total area transferred in this region.

During the research period, deforestation, grass destruction and grain cultivation gradually declined. The area of forest land and grassland reclaimed for agricultural land in the Agricultural Development Region decreased sharply from 462.71 $km^2$ to 13.85 $km^2$; from 2000 to 2005, 50.93 $km^2$ of cultivated land in the Northwest Water Conservation Region reverted back to ecological land. However, at the same time, the area of forest land and grassland transferred for construction and development continued to increase, especially in the ecological functional zone. By the end of the study, the area of this transfer type accounted for 41.97% of the total change area.

*3.2. Spatial Differentiation Characteristics of Landscape Pattern in the Three Functional Regions*

In this paper, first-class land use classification data from the 1980s, 2000 and 2015 was selected to calculate the landscape pattern indicators of the three regions respectively from the class level and landscape level. The spatial and temporal changes were then analyzed.

3.2.1. Landscape Pattern Analysis on the Class-Level

Firstly, looking at the overall levels of the region, the landscape pattern characteristics of the different land-use types vary greatly from one another. Cultivated land makes up the largest proportion of this area. For the duration of the study period, it showed an increasing patch density, decreasing average patch area, and an intensified degree of patch fragmentation. The landscape shape index of grassland is the highest among the six land-use types, and it maintained an irregular shape throughout the study period, indicating that human activities interfere with grassland land types the least. The average patch area of water was higher at the end of monitoring period than at the beginning. Because of the urban expansion of coastal cities such as Tianjin and Tangshan, small areas of water in this region have been integrated, and the development of the agriculture industry provides the possibility for the development of large areas of sea water. The perimeter-area fractal dimension of construction land has the lowest value, and it is the most fragmented and most regular landscape type among all land-use types. The patch density, as well as the average patch area of unused land are the smallest, and both show a downward trend, indicating that the number of broken, unused land patches is gradually increasing.

Secondly, according to the analysis of the dominant land-use types of the different functional regions, significant spatial differences exist among the cultivated land, ecological land, and construction land. The landscape indexes of land-use types in the three functional regions in each period can be found in Appendix A.

In the Agricultural Development Region, the patch density of cultivated land is the smallest, and its average patch area is the largest. Meanwhile, the degrees of combination and aggregation are also the highest. These factors indicate that farmland is the dominant land-use type in this region; the patch density in the ecological area is the largest, and the average patch area is the smallest, suggesting that the degree of fragmentation of cultivated land in the Northwest Water Conservation Region is the highest for the three regions. The arable land patch area in the Beijing–Tianjin Urban Agglomeration decreased the most, and the area was reduced to approximately one-half of the initial stage. Corresponding to this, the area of construction land expanded to more than twice of that in the initial stage.

The patch density of forest land and grassland in the ecological functional region is about 2 to 4 times greater than that of the other two regions. The average patch area of grassland is the largest, and the landscape shape index is also much higher than that of the other land-use types, indicating that grassland in the ecological region is resilient and is the least affected by human activities. Among the three regions, the average patch area of forest land in the living functional area is the largest. This is due to an improvement in the quality of the ecological environment as a result of afforestation, urban green space construction, and other measures taken to improve the vegetation coverage. Therefore, the woodland is distributed in contiguous areas through artificial planning and shows a large area and a sparse distribution of regular plots.

The construction land is most densely distributed in the living area, and most sparsely distributed in the ecological area, with the lowest degree of aggregation; the landscape shape index of construction land in the living area is the lowest, while the perimeter-area fractal dimension is the highest, which indicates that the urban construction area in the Beijing–Tianjin Urban Agglomeration Region has the simplest patch shape but the most complex edge. This is because construction plays a bigger role in providing residence, consumption, leisure, entertainment, and other complex human needs in this region than in the other two regions, and is affected by human activities the most.

### 3.2.2. Analysis of Landscape Patterns on the Landscape-Level

The overall situation of the Beijing–Tianjin–Hebei area is shown in Table 1. By the end of the study period, the patch density increased while the average patch area decreased compared to the initial stage. The landscape shape index and landscape segmentation index continued to rise while the degree indexes of patch aggregation and spread continued to fall, indicating that the overall landscape of the area is highly fragmented. The significant increase in construction caused by urban expansion is the main reason for the exacerbated degree of fragmentation. At the same time, the SHDI and SHEI increased, which shows that the overall landscape heterogeneity has been enhanced. The landscape is developing in a diversified direction, and each patch type is evenly distributed in the landscape of this area, which to a certain extent reflects that the ecological environment of the local city has improved.

During the three observation periods, the patch density increased while the average patch area decreased in the Northwest Water Conservation Region. In addition, the patches became increasingly scattered, the aggregation degree and spread index of patches was the lowest, and the landscape segmentation index was the highest for the three regions, suggesting that the landscape in this area has the worst connectivity and the highest degree of fragmentation compared with the other two regions.

The patch density in the Beijing–Tianjin Urban Agglomeration Region is decreasing, and the average patch area is rising. This implies a high urbanization rate for the two cities and a large-scale increase in the residential area. The landscape shape index is lower than that of the production area and living area, indicating that the land area under artificial planning or reconstruction is larger, causing the geometric shape of the patches to become simple and regular in this region.

The Agricultural Development Region has the highest patch combination degree and spread index but the lowest landscape segmentation index, suggesting that cultivated land is the dominant land type with a compact distribution and a high degree of clustering. The SHDI and SHEI of the landscape are the lowest compared with the other two regions. This indicates that the Agricultural Development Region has the lowest landscape heterogeneity and richness among the three regions, and also reflects that arable land has an absolute advantage among all land-use types in the production functional zone.

**Table 1.** Landscape pattern indexes of the entire Beijing–Tianjin–Hebei area and that of the three functional regions.

| Region | Year | PD | LSI | APA (km²) | PCD (%) | PAD (%) | SI(%) | LSGI | SHDI | SHEI |
|---|---|---|---|---|---|---|---|---|---|---|
| The entire area | 1980s | 0.428 | 245.920 | 233.482 | 99.842 | 89.510 | 51.766 | 0.858 | 1.298 | 0.724 |
| | 2000 | 0.421 | 249.436 | 237.824 | 99.849 | 89.357 | 50.862 | 0.861 | 1.324 | 0.739 |
| | 2015 | 0.460 | 266.072 | 217.453 | 99.829 | 88.664 | 49.069 | 0.895 | 1.361 | 0.760 |
| Northwest Water Conservation Region | 1980s | 0.476 | 208.736 | 210.163 | 99.505 | 85.738 | 48.743 | 0.977 | 1.297 | 0.724 |
| | 2000 | 0.473 | 208.766 | 211.410 | 99.521 | 85.735 | 48.725 | 0.976 | 1.297 | 0.724 |
| | 2015 | 0.516 | 212.277 | 193.668 | 99.472 | 85.496 | 47.852 | 0.978 | 1.315 | 0.734 |
| Beijing-Tianjin Urban Agglomeration Region | 1980s | 0.412 | 73.889 | 242.682 | 99.660 | 91.493 | 51.959 | 0.898 | 1.355 | 0.756 |
| | 2000 | 0.410 | 75.922 | 244.076 | 99.608 | 91.251 | 50.217 | 0.911 | 1.405 | 0.784 |
| | 2015 | 0.381 | 77.857 | 262.575 | 99.496 | 91.070 | 50.263 | 0.940 | 1.399 | 0.781 |
| Agricultural Development Region | 1980s | 0.404 | 132.131 | 247.262 | 99.897 | 91.950 | 61.291 | 0.662 | 1.058 | 0.590 |
| | 2000 | 0.393 | 136.126 | 254.208 | 99.898 | 91.699 | 60.212 | 0.682 | 1.087 | 0.607 |
| | 2015 | 0.446 | 155.927 | 224.270 | 99.889 | 90.491 | 57.479 | 0.734 | 1.142 | 0.637 |

## 4. Discussion

At the 2014 Central Economic Work Conference, China proposed the strategy of the coordinated development of the Beijing–Tianjin–Hebei area for the first time. After a year of research and consultation, on April 30, 2015, the Political Bureau of the CPC Central Committee held a meeting to deliberate and approve the outline of the Beijing-Tianjin-Hebei Coordinated Development Plan. The sustainable development of this area depends on the mutual integration and harmonious utilization of the life community of mountains, rivers, forests, fields, and lakes. At present, the rational utilization and protection of land has become the core principle of the ecological environmental protection of this

area [29]. The spatial-temporal changes of land use over the past 30 years in the Beijing–Tianjin–Hebei collaborative development area were studied using the methods of dynamic degree and transformation matrix of land-use types. The changes of landscape pattern in the area were analyzed based on the class and landscape level.

The study divides the area into three functional areas, namely the Agricultural Development Region, the Beijing–Tianjin Urban Agglomeration Region and the Northwest Water Conservation Region. Due to the influence of natural geographical conditions and economic development level, the main functions of the three sub-regions are different, and the spatial differentiation of contradiction between human and land is also significant. To achieve the development goal of 'intensive and efficient production space, livable living space, and beautiful ecological space', the Beijing–Tianjin–Hebei area should accelerate the implementation of the main functional area strategy and execute spatial-differentiated land resource utilization and protection policies in different sub-regions to optimize the production–ecology–living space pattern and promote sustainable development [30].

The spatial-temporal characteristics of land-use change in the study area can be summarized as follows: multiple dynamic types, complex changes, wide distribution, and large scale. The type of land-use change is mainly the transfer of cultivated land to construction land with a net reduction area of 5217.00 km$^2$, indicating that as the economic strength and the urbanization rate of this region continues to increase, the cultivated land is also being constantly consumed. At present, the government should carefully control the degradation rate of cultivated land, promote the coordinated protection of land resources, and strictly observe the red line of cultivated land to ensure the food security and supply in the whole area. The phenomenon of forest land and grassland occupied by reclamation has been basically controlled. According to the transfer matrix of land-use types, during the 2000–2005 period, 50.93 km$^2$ of cultivated land was transformed into ecological land, suggesting that the project of Returning Farmland to Forest, started in Hebei Province in 2000, has made effective progress [31]. Taking Zhangjiakou City as an example, by the end of 2015, the conversion of farmland to forest reached more than 2757.33 km$^2$, and the greening level improved significantly [32]. According to the monitoring results by the State Forestry and Grassland Administration, the project of Returning Farmland to Forest in Hebei Province conserves 4.916 billion m$^3$ of water per year, and the total mass of windbreak and sand fixation is 102.0759 million tons per year [33]. By returning sloping farmland and low-yield farmland to forests and grassland, land use efficiency has been greatly improved [34], and the functions of water conservation and sand control of forest land and grassland have been fully exerted, which has a high ecological benefit for the entire Beijing–Tianjin–Hebei collaborative development area [31]. Among the three functional areas, the average patch area of forest land in the Beijing–Tianjin Urban Agglomeration Region is the largest, indicating that local green ecological construction started earlier and environmental governance has achieved remarkable results. At present, this area exemplifies the idea that 'urban rate' and 'urban green rate' [34] complement one another and coordinate development; however, we still need to be alert to the phenomenon that primary forests are being replaced by dominant forests which are cultivated artificially, which will reduce the biodiversity and increase the vulnerability of the ecological environment in this area.

From the perspective of landscape level, the patch density in this area increases annually. Meanwhile, the average patch area shows a decreasing trend, and the overall ecological landscape tends to be fragmented. The SHDIs and SHEIs continue to rise, indicating that the heterogeneity of landscape in this area is enhanced. The land-use types are enriched and tend to be evenly distributed, and landscape is developing towards diversification. The findings of this study are consistent with the conclusions of the study in another economic growth pole in the Pearl River Delta urban agglomerations in China, and researchers have found that the degree of land use diversity and that of landscape fragmentation were positively correlated with the degree of urbanization. Similarly, the landscape pattern underwent a fundamental transition from an agricultural-land-use dominant landscape to an urban-land-use dominant landscape in other Chinese cities, in which the urbanization process and spatial range changed significantly [35]. The same trend is happening in other areas around the

world [7]. Dadashpoor et al. [36] analyzes land-use change and its impact on the change in landscape pattern in the Tabriz metropolitan area. Findings revealed that cropland and other ecological lands were the major land-use types transformed into construction land in the process of urban sprawl, and this process has led to increased fragmentation, diversity and reduced aggregation in the studied area.

Overall, the implementation of sustainable practice strategies by the government has resulted in a steady improvement in diversification of the land-use types in the Beijing-Tianjin-Hebei area. However, more stringent practices will be required to ensure that the land in this area develops towards total diversification. We believe that the experiences in ecological environment governance in this area will provide reference for the ecological issues of other urban agglomerations in the world.

This study divided this area into three sub-regions of production, living and ecology, but this is only based on the classification of prefecture-level cities. In the future, we will consider classification based on county-level cities or even higher accuracy to make the results more convincing and credible. In addition, the evidence of data quality and statistical uncertainty of quantitative results should also be added in the future work.

**Author Contributions:** Data preparation X.W., L.Y., F.L. and J.X.; funding acquisition, L.Z.; experiments and original draft preparation, Y.W.; review and editing, X.Z.; supervision, Z.Z. All authors have read and agreed to the published version of the manuscript.

**Funding:** This research was funded by the National Major Science and Technology Project for Water Pollution Control and Treatment (2017ZX0710001).

**Acknowledgments:** The authors would like to thank the anonymous reviewers for their systematic review and valuable comments.

**Conflicts of Interest:** The authors declare no conflict of interest.

## Appendix A

**Table A1.** Landscape indexes of land-use types in the three functional regions in each period.

| Region | Index | Year | Cultivated Land | Forest Land | Grassland | Water Area | Construction Land | Unused Land |
|---|---|---|---|---|---|---|---|---|
| Northwest Water Conservation Region | PD | 1980s | 0.0935 | 0.0888 | 0.1654 | 0.0336 | 0.0853 | 0.0093 |
| | | 2000 | 0.0931 | 0.0869 | 0.1651 | 0.0338 | 0.0847 | 0.0094 |
| | | 2015 | 0.0996 | 0.0901 | 0.1628 | 0.0292 | 0.1245 | 0.0101 |
| | LSI | 1980s | 230.6230 | 170.7003 | 278.9187 | 115.4235 | 100.8453 | 59.4385 |
| | | 2000 | 230.9968 | 170.7693 | 278.8394 | 115.9588 | 99.4215 | 59.8861 |
| | | 2005 | 230.3984 | 170.6857 | 277.1347 | 115.4409 | 98.6337 | 61.0711 |
| | APA (km$^2$) | 1980s | 364.0727 | 386.5052 | 159.843 | 51.1121 | 20.6579 | 187.6345 |
| | | 2000 | 367.2341 | 394.8838 | 158.9228 | 50.1136 | 21.9467 | 180.3992 |
| | | 2015 | 340.1248 | 405.3529 | 142.7108 | 52.1725 | 26.4768 | 153.5581 |
| | PFD | 1980s | 1.5442 | 1.4657 | 1.5273 | 1.6468 | 1.3068 | 1.4781 |
| | | 2000 | 1.5444 | 1.4689 | 1.5319 | 1.6453 | 1.3054 | 1.4776 |
| | | 2015 | 1.5493 | 1.4621 | 1.5267 | 1.6438 | 1.3657 | 1.4464 |
| | PCD (%) | 1980s | 99.7674 | 99.4813 | 98.4621 | 94.6267 | 84.5104 | 97.6214 |
| | | 2000 | 99.7683 | 99.5382 | 98.4558 | 94.8310 | 86.7705 | 97.6426 |
| | | 2015 | 99.7262 | 99.5649 | 98.0775 | 94.8595 | 89.5711 | 97.1053 |
| | PAD (%) | 1980s | 86.3859 | 89.9778 | 81.2993 | 69.7259 | 73.9187 | 84.6479 |
| | | 2000 | 86.3969 | 89.9746 | 81.2356 | 69.3567 | 74.9625 | 84.3407 |
| | | 2015 | 86.0661 | 89.8486 | 80.2114 | 70.3328 | 76.5035 | 83.7294 |

**Table A1.** *Cont.*

| Region | Index | Year | Cultivated Land | Forest Land | Grassland | Water Area | Construction Land | Unused Land |
|---|---|---|---|---|---|---|---|---|
| Beijing-Tianjin Urban Agglomeration Region | PD | 1980s | 0.0402 | 0.0392 | 0.0483 | 0.0623 | 0.2193 | 0.0027 |
| | | 2000 | 0.0425 | 0.0382 | 0.0470 | 0.0716 | 0.2064 | 0.0040 |
| | | 2015 | 0.0577 | 0.0360 | 0.0340 | 0.0517 | 0.2008 | 0.0007 |
| | LSI | 1980s | 77.1804 | 49.5735 | 81.2550 | 53.4376 | 89.9860 | 14.1396 |
| | | 2000 | 81.5742 | 49.9228 | 80.7072 | 55.4462 | 86.3159 | 14.1815 |
| | | 2015 | 93.3216 | 51.3181 | 71.9102 | 57.2759 | 79.6577 | 5.8526 |
| | APA (km$^2$) | 1980s | 1165.479 | 699.8329 | 118.3824 | 127.1666 | 52.2993 | 227.5974 |
| | | 2000 | 1005.497 | 729.6002 | 114.9744 | 122.2250 | 70.0466 | 172.8158 |
| | | 2015 | 597.4391 | 786.2158 | 115.0558 | 142.5139 | 129.0450 | 105.7619 |
| | PFD | 1980s | 1.4843 | 1.4158 | 1.4969 | 1.4618 | 1.3179 | 1.3218 |
| | | 2000 | 1.4875 | 1.4089 | 1.4961 | 1.4483 | 1.3378 | 1.3147 |
| | | 2015 | 1.4879 | 1.4250 | 1.5039 | 1.4847 | 1.3666 | 1.2011 |
| | PCD (%) | 1980s | 99.8670 | 99.6644 | 95.5760 | 98.7875 | 95.0045 | 96.6123 |
| | | 2000 | 99.8023 | 99.7838 | 95.5622 | 98.7715 | 96.6493 | 96.8207 |
| | | 2015 | 99.5932 | 99.7993 | 95.5405 | 99.0427 | 98.6918 | 93.5822 |
| | PAD (%) | 1980s | 93.3686 | 94.4731 | 79.9848 | 88.8882 | 84.3310 | 89.9899 |
| | | 2000 | 92.6605 | 94.4764 | 79.5361 | 89.0177 | 86.6590 | 90.5319 |
| | | 2015 | 90.6690 | 94.3843 | 78.6486 | 87.6817 | 90.8292 | 89.3950 |
| Agricultural Development Region | PD | 1980s | 0.0297 | 0.0375 | 0.0397 | 0.0165 | 0.2783 | 0.0028 |
| | | 2000 | 0.0293 | 0.0376 | 0.0394 | 0.0171 | 0.2674 | 0.0026 |
| | | 2015 | 0.0334 | 0.0452 | 0.0393 | 0.0206 | 0.3068 | 0.0005 |
| | LSI | 1980s | 126.1057 | 108.6783 | 147.7164 | 92.2510 | 192.3988 | 27.0038 |
| | | 2000 | 132.7343 | 108.1388 | 147.0844 | 94.5294 | 187.9331 | 27.4108 |
| | | 2015 | 158.3491 | 117.0194 | 145.4375 | 95.5645 | 216.2612 | 10.9192 |
| | APA (km$^2$) | 1980s | 2296.755 | 204.2056 | 287.6016 | 157.2788 | 34.2610 | 235.8056 |
| | | 2000 | 2262.719 | 203.9046 | 285.6348 | 143.2031 | 43.6619 | 233.2444 |
| | | 2015 | 1871.375 | 203.2601 | 231.2249 | 151.577 | 51.7905 | 317.434 |
| | PFD | 1980s | 1.5297 | 1.4636 | 1.4924 | 1.5736 | 1.2734 | 1.3608 |
| | | 2000 | 1.5321 | 1.4635 | 1.4993 | 1.5769 | 1.2811 | 1.3627 |
| | | 2015 | 1.5298 | 1.4593 | 1.5012 | 1.5848 | 1.3654 | 1.3770 |
| | PCD (%) | 1980s | 99.9739 | 99.0591 | 99.4188 | 98.2160 | 88.8020 | 96.6566 |
| | | 2000 | 99.9739 | 99.0926 | 99.3993 | 98.0155 | 90.9964 | 96.8800 |
| | | 2015 | 99.9702 | 99.2146 | 99.0383 | 98.2361 | 93.6655 | 96.3782 |
| | PAD (%) | 1980s | 95.2838 | 87.8874 | 86.4718 | 82.3287 | 80.6967 | 89.9676 |
| | | 2000 | 94.9661 | 87.9407 | 86.4416 | 81.3472 | 82.9566 | 89.3363 |
| | | 2015 | 93.8217 | 88.1100 | 85.1034 | 83.3449 | 83.2233 | 92.2757 |

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
