# Peer review of "Spatial Differentiation of Land Use and Landscape Pattern Changes in the Beijing–Tianjin–Hebei Area"

_sustainability, doi:10.3390/su12073040_

Round 1

Reviewer 1 Report

Regrettably, after multiple readings I am unable to discern the objectives and primary conclusions of this manuscript, and as such have a very difficult time evaluating the merits. I think this confusion is due to 3 primary reasons:

1) Foremost is a significant need for improvements in English language and style. There are a significant number of minor grammar errors, plus many sentences and paragraphs are difficult to understand.

2) The geographic context and purpose is not clearly explained. The type and nature of the Beijing-Tianjin-Hebei area is not explained for an international audience. The Introduction identifies it as a "collaborative development area", an "economic area", a "political, cultural and technological center", an "urban agglomeration" and a "major national strategy."  All of these are generalized concepts. Is this a formal administrative region defined for unified development planning? Or just a mega-region that has organically developed over time? In addition, this manuscript divides the larger area into 3 sub-regions (Northwest Water Conservation, Bejing-Tianjin Urban Agglomeration, Agricultural development) and 3 functional categories (ecological, production, living), but it is unclear of why and how these two inter-related sub-divisions were conducted. Is dividing the larger region into these sub-regional and functional uses one of the goals of this study? Or are the formal delineated sub-categories of the collaborative development area? To summarize, it is unclear why this region is important for unified study and how and why this region is subdivided. Without understanding this geographic context, all of the land use change and landscape patterns metrics are confusing.

3) The purpose of the spatial analysis is unclear. Calculating land change metrics might be helpful for urban and regional planners, but there is no discussion at all at about how this information could be used to either learn from past policies or inform new policies. The funding source suggests the intent is to address water pollution control and treatment, but likewise there is no connection between landscape change and water resources management. If the purpose of this study is to delineate sub-regions for improved local resource planning and management, then this purpose should be more clearly explained.

I see no reason to doubt the merits of the methodologies, but have 3 suggestions: 1) would like to see some evidence of data quality and statistical uncertainty of quantitative results 2) If dividing the larger region into the 3 sub regions is a major goal of the manuscript, it should be more carefully evaluated on how and why this delineation occurred. 3) the comprehensive dynamic degree of land use metric should have sources given the concept is used in various published studies.

Lastly, there appears to be a healthy literature on remote sensing-based analysis of land change in the Beijing area, but very little is cited here. How does this analysis support or differentiate from conventional understanding of patterns in the region? 

Given all of this broad confusion, I apologize for not having more specific suggestions. As of now, I struggle to see value of this manuscript to an international research audience, either in terms of novel methods or novel insights into urban/regional planning or land change science. The potential is there for providing insights into urban and regional planning, but this manuscript is focused on the analysis more than the application. Significant revision of the manuscript could address this concern.

Reviewer 2 Report

General analysis:

The aims of the article are clear, the methodology is adequate for the purpose of the article, and the structure is functional. However, the presentation of results must be revised, in order to make a clear and fluid text (e.g. text between lines 179 and 188; 252 and 258 must be re-organized and improve structure). The revision of the text will also allow to remove indications associated with a draft version shared by the authors (line 144 "the text continues here"), and allow to eliminate few faults.
The use of tables with a large number of figures must also be considered. The presentation of results on table makes the interpretation difficult. In the case of table 1, wouldn't it be better to use charts as a strategy to show changes for the 7 classes on the three regions?
For the table 2, it should be considered to present the table as an appendix, and use aggregated results for some indexes in order to show trends: fragmentation, diversity, connectivity or others, in order to support text about the main trends.

It would be nice to have a discussion that includes the reference to results about land-use changes in other territories in China.

In detail:

  • line 83 - "classes" instead of "classifications",
  • lines 86 and 87 - in "Materials and Methods" section, what the authors mean by "weft equal area" (line 86) or double standard weft" (line 87)
  • lines 139 and 157 - what the authors mean by "mining residential land"  - it is a specific land use type or two different land uses - mining and residential land ?
  • lines 96 to 98 - problems in the presentation of the formula: formula number 1 is overlapping text,
  • line 169 - area?
  • line 245 - use "land-use types" instead of "land types"
  • in the figure 3, the boundaries/limits for the study area should be kept in each image in order to allow a clear notion of position and proportion,
  • which is the origin of the grasslands in the area - are they natural or derived from land use? Are they associated with specific abiotic conditions?
  • some of the conclusions are not clear: among the measures implemented to control farmland expansion, the authors mention "closing mountains" - what the authors mean by that?

Round 2

Reviewer 1 Report

Thank you for responding to all of the feedback and expanding the explanation and discussion sections. This adequately alleviates my concerns about the initial manuscript, and I now support this manuscript for publication. 

Author Response

Thank you for your systematic review and valuable comments. They are really useful and helpful.

Best wishes to you.

Reviewer 2 Report

All the comments made during the first revision round were considered by the authors. In fact, the revision made by the authors improved the quality of the text and the presentation of results.

"Introduction" and "Discussion" are the two sections of the article that could be improved. Considering the limitation for text, the introduction can be kept in the present form, but discussion, which is focused on results for the study area, can be improved adding ideas about the similarities or differences to other studies in China or in other regions of the world. That information helps readers to understand how far the results are in line, or not, with changes in other territories.

Author Response

  • lines 342to 353 - we added discussion about the similarities to other studies in China and in other regions of the world.

“The findings of this study are consistent with the conclusions of the study in another economic growth pole in the Pearl River Delta urban agglomerations in China, and researchers have found that the degree of land use diversity and that of landscape fragmentation were positively correlated with the degree of urbanization. Similarly, the landscape pattern underwent a fundamental transition from agricultural-land-use dominant landscape to urban-land-use dominant landscape in other Chinese cities in which the urbanization process and spatial range changed significantly[35]. The same trend is happening in other areas around the world[7]. Dadashpoor et al.[36] analyzes land-use change and its impact on the change in landscape pattern in Tabriz metropolitan area. Findings revealed that cropland and other ecological lands were the major land-use types transformed into construction land in the process of urban sprawl, and this process has led to increased fragmentation, diversity and reduced aggregation in the studied area.”

Thank you for your systematic review and valuable comments, they really help us a lot.

Best wishes to you.